# Isolation, Identification and Function of *Pichia anomala* AR_2016_ and Its Effects on the Growth and Health of Weaned Pigs

**DOI:** 10.3390/ani11041179

**Published:** 2021-04-20

**Authors:** Yajun Ma, Zhihong Sun, Yan Zeng, Ping Hu, Weizhong Sun, Yubo Liu, Hong Hu, Zebin Rao, Zhiru Tang

**Affiliations:** 1Key Laboratory for Bio-Feed and Animal Nutrition, College of Animal Science and Technology, Animal Southwest University, Chongqing 400715, China; myj161428@sina.com (Y.M.); sunzh2002cn@aliyun.com (Z.S.); huping0913@163.com (P.H.); swz2012@swu.edu.cn (W.S.); huhong1020@163.com (H.H.); raozebin2020@163.com (Z.R.); 2Fermentation Engineering Department, Hunan Institute of Microbiology, Changsha 410009, China; zengyan1977@163.com (Y.Z.); lyubo123@163.com (Y.L.)

**Keywords:** *Candida utilis*, *Pichia anomala* AR_2016_, isolation and identification, safety evaluation, weaned pigs

## Abstract

**Simple Summary:**

*Pichia anomala* is a simple pseudomycelium with a round cell shape, product killer factors, β-1,3-glucanase, and phytase. *P. anomala* had the functions of anticorrosion, improving the nutritive value of feed, increasing the protein content, and reducing the concentration of nutritive substances. In this study, *P. anomala* AR_2016_ is isolated and identified from solid wine koji, and its culture conditions are optimized. Heat tolerance, bile salt tolerance, and artificial gastric and intestinal juice tolerance are evaluated. In our methodology, thirty weaned pigs were randomly divided into three groups with 10 barrows in each, and fed a maize-soybean meal diet and orally administered 0.85% saline (CK), 1 mL 1 × 10^9^ cfu/mL *Candida utilis* (*C. utilis*), and 1 mL 1 × 10^9^ cfu/mL *P. anomala* once daily for 28 days (*P. anomala*). Our results show that *P. anomala* AR_2016_ grew best in yeast extract peptone dextrose medium with pH 5.0 at 28 °C, 180 r/min and could tolerate 45 °C for 0.5 h, 0.2% pig bile salts, simulated gastric fluid, and 1.0% simulated intestinal fluid. Our study indicates that *P. anomala* AR_2016_ can tolerate high acidity and high bile salts, and has high survivability in the artificial gastric intestinal juice environment. In conclusion, oral administration of *P. anomala* AR_2016_ improves the growth performance, reduces the incidence of diarrhea, enhances intestinal barrier function, and improves microflora in weaned pigs.

**Abstract:**

This study isolates and identifies *Pichia anomala* (*P. anomala*) AR_2016_, and studies its effect on the growth and health of weaned pigs. A *P. anomala* strain from solid wine koji is isolated and identified using 26S rDNA analysis, and its culture conditions are optimized. Heat tolerance, bile salt tolerance, artificial gastric, and intestinal juice tolerance are evaluated. In our methodology, thirty 28 d Large White × Landrace × Rongchang weaned pigs were randomly divided into three groups with 10 barrows in each, and fed a maize-soybean meal diet and orally administered 0.85% saline (CK), 1 mL 1 × 10^9^ cfu/mL *Candida utilis* (*C. utilis*), and 1 mL 1 × 10^9^ cfu/mL *P. anomala* once daily for 28 days. A *P. anomala* strain was identified and named *P. anomala* AR_2016_. *P. anomala* AR_2016_ grew best in yeast extract peptone dextrose medium with pH 5.0 at 28 °C, 180 r/min and could tolerate 45 °C for 0.5 h, 0.2% pig bile salts, simulated gastric fluid, and 1.0% simulated intestinal fluid. Our results show that compared with the CK group, orally administered *P. anomala* AR_2016_ increases average daily gain, the ileal villus height, the ileal mucosal concentrations of occludin and zonula occluens-1, the serum glucose and total protein concentration, total superoxide dismutase, glutathione peroxidase, and total antioxidative capacity activity, the trypsin and lipase activity in jejunal and ileal contents, the jejunal and ileal mucosa mRNA levels of ALP, TNF-α, and TLR-2, and the relative abundance of Bacteroidetes, Actinobacteria, Succinivibrionaceae, Lachnospiraceae, and Prevotellaceae (*p* < 0.05). Compared with the CK group, oral administration of *P. anomala* AR_2016_ decreased the incidence of diarrhea, aspartate aminotransferase activity, alanine amino-transferase-activity, malondialdehyde, D-lactic acid and endotoxin content in serum, the mRNA level of aminopeptidase N of ileum mucosa, and the relative abundance of Proteobacteria, Clostridiaceae, Campylobacteraceae, Vibrionaceae, *Bacillus*, and Pseudon (*p* < 0.05). Collectively, the study indicates that *P. anomala* AR_2016_ can tolerate high acidity and high bile salts, and has high survivability in the artificial gastric intestinal juice environment. Oral administration of *P. anomala* AR_2016_ improves the growth performance, reduces the incidence of diarrhea, enhances intestinal barrier function, and improves microflora in weaned pigs.

## 1. Introduction

After weaning, piglets were fed the diet changed from liquid milk to solid feed, separated from sows, and suffered the changes of the surrounding environment [1]. These stress factors increased the burden of piglets’ gut, changed the morphology and structure of intestinal mucosa, broke the balance of intestinal microbiota, and reduced the immune function and digestion ability [2]. In order to solve the problem of weaning stress, some antibiotic growth promoters were added to the piglets’ diets, but the abuse of antibiotics led to drug resistance, which posed a certain danger to human health and safety. Therefore, many countries restricted or banned the addition of antibiotics to pig diets, and constantly sought safe alternatives to antibiotics.

Yeast is a safe feed additive, and one of the most common probiotics used in pig production. *Pichia anomala* (*P. anomala*), a common strain of Baijiu liquor, is a kind of non-*Saccharomyces cerevisiae* that synthesizes many enzymes to produce esters, higher alcohols, aldehydes, and acids. *P. anomala* secretes killer protein, and is used as a harmless microbial biocontrol agent in feed preservation or storage [3,4,5]. *P. anomala* has the characteristics of increasing feed protein content, antifungal activity, promoting animal weight gain, and reducing animal mortality, but few reports have been reported on pig research. Olstorpe et al. reported that inoculation of *P. anomala* in cereal feed could improve feed nutritional value, increase protein content and reduce the concentration of antinutrients by reducing mold and Enterobacteriaceae [6]. There was a large amount of phytase outside the cell of *P. anomala*, which can improve the utilization rate of phosphate in animals [7]. Tayel et al. found that ectochitinase and β-1,3-glucanase produced by *P. anomala* could inhibit the growth of Aspergillus flavus, and had antifungal activity [8].

*P. anomala*, a milky white colony, was a simple pseudomycelium with a round cell shape, which sprouts and reproduces vegetative somatic cells [3]. Supplement of *P. anomala* cells in diet could increase protein content in feed and improved animal weight gain. Supplement of *P. anomala* cells in diet could significantly reduce mortality of animals. *P. anomala* was resistant to Candida albicans, gastric acid, and bile salt. Its liquid and solid fermentation products also contain killer factors, β-1,3-glucanase and phytase [9,10,11]. Previous studies by our research group had shown *P. anomala* to be rich in various essential amino acids and bioactive substances with high safety [12]. In this study, *P. anomala* was isolated and identified from traditional solid wine koji, and the effects of *P. anomala* on the growth and health of weaned pigs were investigated to provide data for the application of *P. anomala* in weaned piglets.

## 2. Materials and Methods

### 2.1. The Isolation, Identification, Heat Tolerance, Bile Salt Tolerance, and Artificial Gastric and Intestinal Juice Tolerance of P. anomala AR_2016_

#### 2.1.1. Microorganisms, Culture Media, and Reagents

Traditional solid wine koji with yeast was sampled from Anren County, Hunan province, China.

Enrichment culture medium comprised 5% (*w*/*v*) glucose, 0.05% (*w*/*v*) yeast extract, 0.1% (*w*/*v*) ammonium sulfate, 0.01% (*w*/*v*) ferric sulfate heptahydrate, 0.25% (*w*/*v*) potassium dihydrogen phosphate, 0.1% (*w*/*v*) urea, and 0.003% (*w*/*v*) bengal red, was adjusted pH to 5.0 and sterilized for 15 min at 115 °C.

The PDA medium comprised 200.0 g potato, 20.0 g glucose, 20.0 g agar powder, distilled water. A constant volume of 1 L was kept at natural pH and sterilized for 15 min at 115 °C.

YPD culture medium comprised 2% (*w*/*v*) glucose, 1% (*w*/*v*) yeast powder, 2% (*w*/*v*) peptone, 2% (*w*/*v*) agar, and was adjusted to pH 6.0 and sterilized for 15 min at 115 °C.

The solid toxin culture medium comprised 2% (*w*/*v*) glucose, 1% (*w*/*v*) yeast powder, 2% (*w*/*v*) peptone, 2% (*w*/*v*) agar. *P. anomala* cells in liquid culture were cultivated in YPD broth supplemented with 5% (*v*/*v*) glycerol which was adjusted to pH 4.5 with 100 mM citrate phosphate buffer.

#### 2.1.2. Isolation and 26S rDNA D1/D2 Identification of *P. anomala*

One gram of solid wine koji with yeast was mixed with 100 mL yeast liquid enrichment medium. The 1 mL suspension was serially diluted 10 times to 10^−6^. The 100 μL suspension from diluted samples was plated on PDA solid medium and cultured for 28 °C for 2–3 d. Three-day-old cultures in PDA agar were used for cellular and colony morphology analysis of yeast. Cell morphology of *P. anomala* was photographed using a microscope.

Three-day-old cultures of yeast in PDA liquid medium were collected and directly used for DNA extraction according to the instructions of the Power DNA Isolation Kit (Mobio, Carlsbad, CA, USA). DNA was quantified using a Nanodrop spectrophotometer (Nyxor Biotech, Paris, France) following staining using the Quant Pico Green dsDNA Kit (Invitrogen Ltd., Pailsey, UK). PCR amplification of the D1/D2 region of Yeast 26S rDNA was performed using universal primers NL1 (5′-GCATATCAATAAGCG GAGGAAAAG-3′) and NL4 (5′-GGTCCGTGTT TCAAGACGG-3′) [13]. The cycling parameters were as follows: 4 min initial denaturation at 94 °C; 25 cycles of denaturation at 94 °C (40 s), annealing at 55 °C (45 s), elongation at 72 °C (30 s); and final extension at 72 °C for 10 min. PCR reaction system: DNA Template (20–50 ng/μL) 0.5 μL; 10 × Buffer (with Mg^2+^) 2.5 μL; dNTP (2.5 mM) 1 μL; Taq polymerase 0.2 μL; primer F (10 μM) 0.5μL; primer R (10 μM) 0.5 μL; ddH_2_O 25 μL. The PCR products were separated by 1.0% agarose gel electrophoresis (150 V, 100 mA, 20 min) and purified by using the QIAquick Gel extraction kit (Qiagen, Dusseldorf, Germany). PCR product was sequenced by Sangon Biotech Co. Ltd., Shanghai, China. According to the base sequence of the yeast, the detailed yeast was found in the NCBI gene bank, the phylogenetic tree was constructed, and the species of the yeast was determined by the base sequence of the yeast. BLAST analysis was carried out to obtain the known strains with high homology with yeast 26S rDNA sequence, and the gene sequences of the strains with high similarity were obtained from the GenBank library for the establishment of a phylogenetic tree. This yeast was named *P. anomala* AR_2016_ and was stored in China Center for Type Culture Collection (No. CCTCC M2017594, http://cctcc.whu.edu.cn/, accessed on 20 October 2017).

#### 2.1.3. Anti *Candida Albicans* Test

*Candida albicans* (CMCC: 98001) was plated evenly on the solid toxin culture medium. After *C. albicans* liquid was dried, *P. anomala* AR_2016_ was inoculated on the tablet and cultured for 2–3 d at 28 °C. If there was a transparent circle around *P. anomala* AR_2016_, it was evident that *P. anomala* AR_2016_ had killer activity [14].

#### 2.1.4. Optimization of Culture Conditions for Strain AR_2016_

A single colony of *P. anomala* AR_2016_ was inoculated into YPD liquid medium and cultured for 24 h at 28 °C and 180 r/min. The seed culture of *P. anomala* AR_2016_ was inoculated by placing 2.5% (*v*/*v*) in YPD liquid medium at a pH of 3.0, 4.0, 4.5, 5.0, 6.0, or 7.0, and culturing at 28 °C, at 180 r/min. Each group had three replicates. The OD_600_ of *P. anomala* AR_2016_ cultures were determined at 1 h, 3 h, 5 h, 7 h, 9 h, 11 h, 13 h, 15 h, 17 h, 19 h, 21 h, 23 h, 28 h, 42 h, and 48 h.

The seed culture of *P. anomala* AR_2016_ were inoculated by 2.5% (*v*/*v*) inoculating amount in YPD liquid medium with 5.0 pH and cultured at 180 r/min and 20, 25, 28, or 33 °C, respectively. Each group had 3 replicates. The OD_600_ of *P. anomala* AR_2016_ cultures were determined at 1 h, 6 h, 11 h, 16 h, 21 h, 26 h, 31 h, 36 h, 41 h, and 47 h.

The seed culture of *P. anomala* AR_2016_ were inoculated by 2.5% (*v*/*v*) inoculating amount in YPD liquid medium with 5.0 pH, respectively, and cultured at 120, 150, 180, 210 r/min, and 28 °C, respectively. Each group had 3 replicates. The OD_600_ of *P. anomala* AR_2016_ cultures were determined at 1 h, 6 h, 11 h, 16 h, 21 h, 26 h, 31 h, 36 h, 41 h, and 47 h.

#### 2.1.5. Resistance Testing of Strain *P. anomala* AR_2016_

Logarithmic phase *P. anomala* AR_2016_ cells were centrifuged and washed twice with 0.85% [12], then resuspended to 1 × l0^7^ cfu/mL in 0.85% saline and immersed in a water bath at 28, 35, 45, 55, 65, 75, 85, 95 °C for 0.5 h, then cooled rapidly and diluted from 10^−1^ to 10^−6^ times, cultured at 28 °C for 2–3 d and the colony forming units (CFU) number was then counted. Each group had three replicates.

The logarithmic phase *P. anomala* AR_2016_ cells were inoculated using 10% (*v*/*v*) in YPD liquid medium which included 0.0%, 0.1%, 0.2%, 0.3%, 0.4%, or 0.5% of a pig bile salt. Plates were then cultured at 28 °C for 4 h, then diluted from 10^−1^ to 10^−6^ times, and cultured at 28 °C for 2–3 d, when the CFU was counted. Each group had three replicates.

The logarithmic phase *P. anomala* AR_2016_ was inoculated using 10% (*v*/*v*) in artificial gastric juice and cultured at 28 °C for 0, 0.5, 1, 2, 2.5, 2.5, 3 or 3.5 h, and then diluted from 10^−1^ to 10^−6^ times, cultured at 28 °C for 2–3 d, when the CFU number was counted. Each group had three replicates.

The logarithmic phase *P. anomala* AR_2016_ was inoculated using 10% (*v*/*v*) in artificial intestinal fluid and cultured at 28 °C for 0, 2, 4, 6, 8 or 10 h, then diluted from 10^−1^ to 10^−6^ times, cultured at 28 °C for 2–3 d and the CFU number was then counted. Each group had three replicates.

### 2.2. Experimental Design and Dietary Composition for Pigs

#### 2.2.1. Microorganisms and Reagents

*Candida utilis* was purchased from the Deutsche Sammlung von Mikroorganismen und Zellkulturen (No. DSM 2361). Total superoxide dismutase (SOD), total antioxidant capacity, aminotransferase aspartate, alanine aminotransferase, glutathione peroxidase, glucose, total cholesterol, total protein, trypsin, amylase, and lipase were purchased from Nanjing Jiancheng Bioengineering Institute (Nanjing, China). Rabbit-actin, occludin, ZO-1, β-defensin-2, and goat-rabbit all purchased from Cell Signaling Technology (Danvers, America). Seal fluid and developer were purchased from Millipore. AMV First Strand cDNA Synthesis Kit was purchased from Shanghai Bioengineering (Shanghai, China). SYBR^®^ Green PCR Kit and PowerFecal DNA Isolation Kit was purchased from Germany Qiagen reagent company (Dusseldorf, Germany).

#### 2.2.2. Use and Care of Pigs

Thirty, 28 d old, Large White × Landrace × Rongchang weaned pigs (7.48 ± 0.22 kg) were randomly divided into three groups of 10 barrows each, and were orally administered 1 mL of 0.85% saline (CK), 1 mL 1 × 10^9^ cfu/mL *C. utilis* in 0.85% saline (*C. utilis*), or 1 mL 1 × 10^9^ cfu/mL *P. anomala* in 0.85% saline (*P. anomala*) once daily (12:00) for 28 days. All pigs were fed a diet formulated according to the National Research Council requirements (2012). The ingredients and composition of the basal diet are given in Table 1. The pigs were kept individually in pens (1.5 m length × 0.5 m width × 0.8 m depth) in a mechanically ventilated and temperature-controlled room (22 ± 1.2 °C). Feed and water were available on an ad libitum basis. All experimental procedures were approved by the License of Experimental Animals (SYXK 2014-0002) of the Animal Experimentation Ethics Committee of Southwest University, Chongqing, China.

#### 2.2.3. Growth Performance and Incidence of Diarrhea

Feed intake and incidence of diarrhea in pigs were recorded daily. The pigs were weighed on day 29 prior to the morning feed. The average daily feed intake, average daily weight gain, and feed conversion ratio (FCR) were calculated according to the following formula:average daily weight gain (g/d) = (final weight − initial weight) (g) ÷ 28 (d),(1)
average daily food intake (g/d) = total food intake (g) ÷ 28 (d),(2)
feed conversion rate = average daily weight gain (g) ÷ average daily food intake (g),(3)
Diarrhea incidence (%) = the number of pigs with diarrhea ÷ (the number of pigs × test days) × 100%.(4)

#### 2.2.4. Measurements and Sampling

Prior to the morning feed on day 29, five piglets were selected from each group, and a 10 mL blood sample was collected from piglets. The blood sample was undisturbed for 60 min and then centrifuged at 3500× *g* for 10 min at 4 °C to harvest the serum. Serum was stored at −20 °C for biochemical analysis and enzyme linked immunosorbent assay (ELISA).

After blood sampling, the five pigs with a similar average weight selected from each group were anesthetized with an intravenous injection of sodium pentobarbital (50 mg/kg Basal body weight) and bled by exsanguination. About 2 cm of jejunal and ileal tissue was excised from the midpoint of the jejunum, gently rinsed using cold saline, and placed into 10% formalin solution for 24 h before hematoxylin and eosin (H&E) staining. The jejunal and ileal mucosa were rinsed with cold saline, and the mucosa was scraped gently with a scalpel blade and collected. The harvested jejunal mucosa was immediately frozen in liquid N_2_ and stored at −80 °C for real-time polymerase chain reaction (PCR).

#### 2.2.5. Serum Biochemical Index Analysis

The presence of total superoxide dismutase (SOD) and total antioxidant capacity (T-AOC), aspartate aminotransferase (AST), alanine aminotransferase (ALT), glutathione peroxidase (GSH-px), glucose (Glu), total cholesterol (T-CHO), total protein (TP) in serum, and the presence of trypsin (TRS), amylase (AMS), and lipase (LPS) in the jejunum and ileum were determined using colorimetric methods with a reagent kit according to the manufacturer’s instructions (Nanjin Jianchen Institute of Bioengineering, Nanjing, Jiangsu, China).

#### 2.2.6. Hematoxylin-Eosin Staining

The morphology of the ileum was analyzed by H&E staining as described by Wang et al. [15]. Sliced samples were viewed under an optical microscope (Carl Zeiss Inc., Oberkochen, Bayern, Germany). Five pictures were taken of each sample, and five fields in each picture were used to analyze villus height and crypt depth using image analysis software (Intronic GmbH and Co., Rothenstein, Berlin, Germany).

#### 2.2.7. Western Blot Analysis

About 100 mg of the jejunal and ileal mucosa was homogenized in 1 mL RIPA buffer (50 mM Tris-base, 1.0 mM ethylene diamine tetraacetic acid (EDTA), 150 mM NaCl, 0.1% sodium dodecyl sulfate (SDS), 1% Tritox-100, 1% sodium deoxycholate, and 1 mM phenylmethylsulfonyl fluoride (PMSF)) and separated by SDS-polyacrylamide gel electrophoresis (SDS-PAGE). The proteins were transferred to a polyvinylidene fluoride (PVDF) membrane by the semi-dry transfer method. The PVDF membranes were blocked in a blocking buffer overnight at 4 °C, then incubated in blocking buffer with rabbit-anti-rat BD-2 (ab178728, 1:1000, Abcam, Cambridge, MA, USA), β-actin (5125S, 1:1000, CST, Danvers, MA, USA), occludin (ab31721; 1:1000, Abcam) or zonula occludens (ZO)-1 (ab216880, 1:1000, Abcam), and incubated in blocking buffer with F(ab)_2_ of goat-anti-rabbit Ig (FAB127288, 1: 2500, Fantibody) labeled with horseradish peroxidase and with diluted in phosphate-buffered saline solution. The PVDF membrane was soaked in a chemiluminescent liquid (Millipore, Boston, MA, USA). Pictures were taken using a Chemiluminescence Imaging System (Bio-Rad, Bio rad Hercules, CA, USA).

#### 2.2.8. Real-Time Reverse Transcription-Polymerase Chain Reaction

Frozen jejunal and ileal mucosa samples (50 mg) were homogenized in 5 mL TRIzol reagent (Invitrogen, Carlsbad, CA, USA) containing RNAlater (Invitrogen, Carlsbad, CA, USA), and total RNA was isolated according to the manufacturer’s recommendations. Total RNA was reverse transcribed to cDNA using an AMV First Strand cDNA Synthesis kit (Bio Basic Inc., Markham, ON, Canada).

Glyceraldehyde-3-phosphate dehydrogenase (GAPDH) was tested as an adequate housekeeping gene. Real-time PCR analysis was performed on GAPDH, alkaline phosphatase (ALP), aminopeptidase N (APN), toll-like receptors 2 (TLR-2), tumor necrosis factor-α (TNF-α), and interleukin-10 (IL-10) mRNAs in the jejunal and ileal mucosa. Primers for genes in the jejunal and ileal mucosa are shown in Table 2. PCR analysis was performed using the SYBR Green method and the ABI 7900 Sequence Detection System (Applied Biosystems, Foster, CA, USA). Thermal cycling parameters were set as follows: 94 °C for 30 s, followed by 40 cycles at 94 °C for 5 s, corresponding annealing temperature (GAPDH 55 °C, ALP 55 °C, APN 55 °C, TLR-2 50 °C, TNF-α 55 °C, L-10 50 °C) for 20 s, and then 72 °C for 20 s. The PCR products were identified by melting curve analysis and sequencing (Sangon Biotech, Shanghai, China). Standard calibration curves of target genes were made according to the cDNA concentration and Ct value. The expression quantity of the target gene was estimated by the calibration curve and normalized against the expression of GAPDH using REST 2009 Software.

#### 2.2.9. Cecum Microflora Sequencing

The total DNA of cecum contents was extracted with the MOBIO PowerFecal DNA Isolation Kit, and the DNA purity was detected. The DNA purity was evaluated by OD_260/280_, and the required purity was between 1.7 and 1.9. The extracted total DNA was sent to Chengdu Luoning Biological Technology Co for sequencing and analysis of microbial 16S rDNA fragments. Briefly: After the sample DNA was purified, the specific primers with Barcode were synthesized to amplify the 16S rDNA V4 region of the sample. The primers were 515F (5′-GTGCC AGCMG CCGCG GTAA-3′) and 806R (5′-GGACT ACHV GGGTW TCTAAT-3′). Three replicates were performed for each sample, and the recovered products of PCR were detected and quantified with Qubit 2.0. Then, according to the sequencing quantity requirements of each sample, the corresponding proportion was mixed. Illumina’s TruSeq DNA PCR-Free Sample Prep Kit was used to construct the library, and Illumina’s MiSeq Reagent Kit v2 was used for MiSeq sequencing. PE reads were obtained from Miseq sequencing were spliced with FLASH (https://ccb.jhu.edu/software /FLASH/, accessed on 30 October 2017). At first, and the quality of the sequence was controlled at the same time. Data filtering was completed after the removal of low-quality bases and contaminated sequences of joints. The samples were then analyzed with UPARSE [16].

#### 2.2.10. Determination of Microbial Decarboxylase in the Jejunum and Ileum

Samples of jejunal and ileal contents of pigs were collected, centrifuged for 5 min at 10,000 rpm, and the supernatant was poured out. After repeated washing and precipitation with normal saline three times, the supernatant was centrifuged for 5 min at 10,000 rpm again. The sample was diluted with normal saline until the OD_600_ was 0.8. Then, 10 mL was centrifuged for 5 min at 10,000 rpm. The supernatant was poured out, and 500 μL of physiological saline was added, together with centrifuge tube ice for 30 s. Ultrasonic processing was repeated three times for cell breakage. Following centrifugation at 4 °C for 15 min, at 16,000× *g*, the supernatant was placed into a new pre-cooled 1.5 mL centrifuge tube for microbial decarboxylase determination according to the method of Tang et al. [14].

### 2.3. Data Calculation and Statistical Analysis

All data are presented as means ± Standard error of the mean (SEM). The experimental data were prepared by Excel 2016 software. The data were subjected to one-way ANOVA analysis in SAS statistical software 8.1 (SAS Institute, Inc. Cary, NC, USA), according to a completely randomized one-factorial design. Duncan’s test was performed to identify differences among groups. Significance was set at *p* < 0.05.

## 3. Results

This section may be divided by subheadings. It should provide a concise and precise description of the experimental results, their interpretation, as well as the experimental conclusions that can be drawn.

### 3.1. Optimal Culture of P. anomala AR_2016_

#### 3.1.1. The Colony Characteristics of the Strain *P. anomala* AR_2016_

*P. anomala* AR_2016_ was isolated, identified, and stored in the China Center for Type Culture Collection (No. CCTCC M2017594). The colonies of strain AR_2016_ were milky white, dry, dull, with an opaque surface (Figure 1A). The center of the colony was convex, and the edge was rough. The diameter of the colony was about 3 mm. A single colony of strain AR_2016_ was easily selected from the solid culture medium. Strain AR_2016_ was observed under the microscope, and its cells were round, budding and reproductive, single and dispersed (Figure 1B). The results of the 26S rDNA analysis confirmed *P. anomala*.

#### 3.1.2. Physiological and Biochemical Characteristics of AR_2016_

The carbon source assimilation results of AR_2016_ showed that strain AR_2016_ could assimilate galactose, sucrose, fiber disaccharide, sorbitol, melezitose, inositol, trehalose, acetamide, mannitol, rhamnose, xylose, ribose, glycerin, and arabinose other carbon, but could not assimilate melitose and raffinose (Table 3). According to the standard data of yeast identification manual, it was determined that strain AR_2016_ was consistent with the standard strain of *P. anomala*.

The conserved internal transcribed sequence (ITS) region of the yeasts was amplified by PCR and submitted sequencing data in NCBI genbank (GenBank accession number is MT940527). The full length of the 26S rDNA sequence of AR_2016_ was 596 bp. A BLASTN (http://www.blast.ncbi.nlm.nih.gov/Blast, accessed on 11 November 2020) search of the sequenced PCR products revealed that the isolated yeast strains were *P. anomala* with a 99% identity and coverage. Phylogenetic trees of this yeast with high homology were selected, and MEGA5.05 software was used to construct strain AR_2016_ (Figure 2A). The killing activity of *P. anomala* AR_2016_ was detected (Figure 2B). Transparent circles appeared around *P. anomala* AR_2016_ strain after 48 h of culture at 20 °C, which proved that *P. anomala* AR_2016_ had inhibitory effects on *C. albicans*.

#### 3.1.3. Optimization of Culture Conditions for AR_2016_

The initial pH of the culture medium affects the life activities and material metabolism of microorganisms (Figure 3A). *P. anomala* AR_2016_ cultured at pH 3 takes more time to enter the logarithmic phase at pH 4.0, 4.5, 5.0, 6.0, or 7.0. When *P. anomala* AR_2016_ was cultured at pH 4.0, 4.5, 5.0, 6.0, and 7.0, the growth of this yeast was relatively slow in the first 4 h, and then entered the logarithmic phase after 4 h, which lasted until 18 h. The optimal initial culture medium pH of AR_2016_ was 5.0.

Temperature has important effects on the growth of microorganisms (Figure 3B). The strain AR_2016_ can grow well at 20–32 °C and has a strong survival ability. After entering the stable period, the OD_600_ of AR_2016_ at 28 °C was the largest. The optimal growth temperature of AR_2016_ was 28 °C.

The AR_2016_ strain entered the logarithmic phase after 4 h at different rotating speeds (Figure 3C). The growth rate of AR_2016_ was highest when the rotation speed was 180 rpm.

#### 3.1.4. The Resistance of AR_2016_

When AR_2016_ was treated at 35 and 45 °C for 0.5 h, there was no significant change in the number of living cells (*p* > 0.05). When treated at 55 °C for 0.5 h, the number of living cells of AR_2016_ was decreased (*p* < 0.05). When treated at 65 °C for 0.5 h, the AR_2016_ was almost all dead (Figure 4A).

AR_2016_ treated with 0.1%, and 0.2% pig bile salts for 4 h did not affect the living cell numbers (*p* > 0.05) (Figure 4B). AR_2016_ treated with 0.3%, 0.4%, and 0.5% pig bile salts for 4 h, decreased AR_2016_ living cell numbers (*p* < 0.05), but there were still 50% live cells. AR_2016_ treated with 0.5% pig bile salts for 4 h decreased AR_2016_ living cell numbers sharply (*p* < 0.05), leaving only 10% live cells.

AR_2016_ treated with artificial gastric juice at pH 2.0 for 0.5 and 1 h had no effects on living cells (*p* > 0.05) (Figure 4C). AR_2016_ treated with artificial gastric juice at pH 2.0 for 1.5 h, decreased living cell numbers (*p* < 0.05). With the extension of treatment time, the number of living cells decreased continuously, but there were still living bacteria after treatment for 3 h. The results showed that the artificial gastric juice had a certain inhibitory effect on the growth of *P. anomala* AR_2016_, and the strain also had a certain tolerance to gastric acid.

AR_2016_ treated with the artificial intestinal fluid for 2 and 4 h showed no significant change (*p >* 0.05) (Figure 4D). Compared with AR_2016_ in artificial intestinal fluid for 6 h, AR_2016_ treated for 8 h increased (*p* < 0.05). The results showed that *P. anomala* AR_2016_ can grow and reproduce in the artificial intestinal fluid and pass through the intestinal environment smoothly.

### 3.2. Effects of *P. anomala* AR_2016_ on Growth and Health of Weaned Pigs

#### 3.2.1. Growth Performance and Diarrhea Incidence

Average daily gain, average daily food intake (*p* < 0.05), and FCR of pigs differed among the three groups (*p* < 0.001) (Table 4). Pigs orally administered *C. utilis* and *P. anomala* had higher average daily gain and average daily food intake (*p* < 0.05), and lower FCR (*p* < 0.05) than pigs fed the basal diet. However, average daily gain, average daily food intake, and FCE did not differ between *C. utilis* and *P. anomala* groups (*p* > 0.05). Diarrhea incidence on days 7, 14, 21, and 28 d were different among the three treatments (*p* < 0.001), i.e., lowest in the *P. anomala* group and highest in CK group (*p* < 0.05) (Figure 5).

#### 3.2.2. Plasma Biochemical Index

Serum glucose, total protein, total superoxide dismutase, total antioxidative capacity, glutathione peroxidase, malondialdehyde, D-lactate, and endothelin-1 concentration of pigs differed among the three groups (*p* < 0.001) (Table 5). Pigs orally administered *C. utilis* and *P. anomala* had higher serum glucose and total protein concentration (*p* < 0.05) than pigs fed the basal diet. Pigs orally administered *C. utilis* had higher serum glucose and total protein concentration (*p* < 0.05) than those orally administered *P. anomala*. Pigs orally administered *C. utilis* and *P. anomala* had higher serum total superoxide dismutase, total antioxidative capacity, and glutathione peroxidase concentration (*p* < 0.05) than pigs fed the basal diet. Pigs orally administered with *C. utilis* had a lower serum total superoxide dismutase, total antioxidative capacity, and glutathione peroxidase concentration (*p* < 0.05) than those orally administered *P. anomala*. Pigs orally administered *C. utilis* and *P. anomala* had lower serum Malondialdehyde, D-lactate, and endothelin-1 concentration (*p* < 0.05) than pigs fed the basal diet.

#### 3.2.3. Digestive Enzyme Activity of Weaned Pigs

The trypsin, amylase, and lipase activity of jejunal contents in the *C. utilis* group and the *P. anomala* group were higher than those in CK group (*p* < 0.05), and the trypsin and amylase activity of the *P. anomala* group were higher than those in the *C. utilis* group (*p* < 0.05) (Table 6). The lipase activity of the *C. utilis* group was higher than that of *P. anomala* group (*p* < 0.05). The trypsin activity and lipase activity of the ileal contents in the *C. utilis* and *P. anomala* groups were higher than those in CK group (*p* < 0.05), but there was no difference in trypsin activity between the *C. utilis* group and *P. anomala* group (*p* < 0.05). The lipase activity in the *C. utilis* group was higher than that in the *P. anomala* group (*p* < 0.05). There was no difference in amylase activity among the three groups (*p* > 0.05).

#### 3.2.4. Structure of Ileal Mucosa of Weaned Pigs

The height of villi in the *P. anomala* group was higher than that in the CK group and the *C. utilis* group (*p* < 0.05), and there was no difference between the CK and *C. utilis* groups (*p* > 0.05) (Table 6). The crypt depth of the *C. utilis* group was lower than that in the CK group and the *P. anomala* group (*p* < 0.05). There were no differences between the CK group and the *P. anomala* group (*p* > 0.05). There was a difference in the villi height/crypt depth (V/C) ratio among the three groups, i.e., greatest in the *P. anomala* group and lowest in the CK group (*p* < 0.01). Compared to the CK group, the ileum villi of the *C. utilis* and *P. anomala* groups had clear contours, a neat and dense arrangement, and fewer broken fragments, while the ileal villi of the CK group have severe ruptures, deletions, with some cells falling off and becoming necrotic (Figure 6).

#### 3.2.5. Occludin, ZO-1, and β-defensin-2 Concentrations in Intestinal Mucosa

The occludin and ZO-1 concentrations in the jejunal mucosa and the β-defensin-2 concentration in the jejunal and ileal mucosa in the three groups didn’t differ (Figure 7). The ileal mucosa ZO-1 concentrations differed among the three groups (*p* < 0.05), i.e., were greatest in the *P. anomala* group and lowest in the CK group (*p* < 0.01). The occludin concentration differed among the three groups (*p* < 0.05), i.e., greater in the *C. utilis* and *P. anomala* groups than in the CK group (*p* < 0.01). There was no difference in ileal mucosa occludin concentration in the *C. utilis* and *P. anomala* groups (*p* > 0.05). There was no difference in ileal mucosa β-defensin-2 concentration among the three groups (*p* > 0.05).

#### 3.2.6. The mRNA Level of ALP, APN, TLR-2, TNF-α, and IL-10 in the Intestinal Mucosa

The mRNA levels of ALP in jejunal and ileal mucosa in the *C. utilis* and the *P. anomala* group were higher than those in the CK group (*p* < 0.05) (Table 7). There was no difference in the mRNA levels of ALP in the jejunal and ileal mucosa in the *C. utilis* and the *P. anomala* groups (*p* > 0.05). The ileal mucosa APN mRNA level in *P. anomala* and *C. utilis* groups was lower than that in the CK group (*p* < 0.05). There was no difference in the ileal mucosa APN in the *P. anomala* and *C. utilis* groups (*p* > 0.05). The TLR-2 mRNA level in the jejunal mucosa in the *P. anomala* group was higher than that in the CK group and the *C. utilis* group (*p* < 0.05), and there was no difference in the TLR-2 mRNA level in the ileal mucosa between the CK group and the *C. utilis* group (*p* > 0.05). There was a difference in the TNF-α mRNA levels in the jejunal and ileal mucosa in the three groups, i.e., greatest in the *C. utilis* group and lowest in the CK group (*p* < 0.01).

#### 3.2.7. Microbial Amino Acids Decarboxylase Activity

There was no difference in arginine decarboxylase activity, lysine decarboxylase (LDC) activity, methionine decarboxylase activity, and tryptophan decarboxylase activities in jejunal contents among the three groups (*p* > 0.05) (Table 8). There was no difference in the arginine decarboxylase activity and methionine decarboxylase activity in the ileum contents among the three groups (*p* > 0.05). The microbial histidine decarboxylase (HDC) activity in jejunal and ileal contents in the *C. utilis* group and the *P. anomala* group was lower than that in the CK group (*p* < 0.05), and there was no difference in the histidine decarboxylase activity in the jejunal and ileal contents between the *C. utilis* group and the *P. anomala* group (*p* > 0.05). There was a difference in the lysine decarboxylase activity and tryptophan decarboxylase (TDC) activity of ileal contents among the three groups, i.e., greatest in the CK group and lowest in the *P. anomala* group (*p* < 0.01).

#### 3.2.8. Microflora of Cecum of Weaned Pigs

There was a difference in the Observed index, Chao1 index, and PD index of the microflora of the cecum of weaned pigs in the three groups, i.e., greatest in the *P. anomala* group and lowest in the CK group (*p* < 0.01) (Figure 8A). There was a difference in the relative abundance of *Peseudon* in the cecum of weaned pigs in the three groups, i.e., greatest in the CK group and lowest in the *C. utilis* group (*p* < 0.01). There was a difference in the relative abundance of Prevotellaceae in the cecum of weaned pigs in the three groups, i.e., greatest in the *C. utilis* group and lowest in the CK and *P. anomala* groups (*p* < 0.01). There was a difference in the relative abundance of Rhodospirillales in the cecum of weaned pigs among the three groups, i.e., greatest in the *P. anomala* group and lowest in the CK group (*p* < 0.01). There was a difference in the relative abundance of *Bacillus* spp. in the cecum of weaned pigs, i.e., the greatest in the *P. anomala* group and the lowest in the CK group (*p* < 0.01).

There was no difference in populations of *Prevotellaceae*, *Firmicutes*, *Bacteroidaceae*, *Ruminococcaceae*, *Veillonellaceae*, *Acidaminococcaceae*, and *Enterobacteriaceae* in the cecum of weaned pigs in the three groups (*p* < 0.01) (Table 9). There was a difference in *Bacteroidetes*, *Proteobacteria*, *Lachnospiraceae*, *Actinobacteria*, *Clostridiaceae*, *Lactobacillaceae*, *Succinivibrionaceae*, *Desulfovibrionaceae*, and *Campylobacteraceae* in the cecum of weaned pigs in the three groups (*p* < 0.01). Pigs oral administrated *P. anomala* could increase populations of *Bacteroidetes*, *Lachnospiraceae*, and *Succinivibrionaceae* in their cecum (*p* < 0.05). Pigs oral administrated *P. anomala* could increase populations of *Proteobacteria*, *Actinobacteria*, *Clostridiaceae*, *Lactobacillaceae*, *Desulfovibrionaceae*, and *Campylobacteraceae* in the cecum (*p* < 0.05).

## 4. Discussion

Authors should discuss the results and how they can be interpreted from the perspective of previous studies and of the working hypotheses. The findings and their implications should be discussed in the broadest context possible. Future research directions may also be highlighted.

*P. anomala* AR_2016_ was isolated from traditional wine and medicine in Anren County, Hunan province, China, and identified by 26S rDNA sequencing. *P. anomala* AR_2016_ contains 59.56% crude protein and various amino acids, with a high content of aspartic acid, glutamate, and glycine [12]. AR_2016_ produces a killer toxin (protein or glycoprotein toxin) that can be used to control harmful microorganisms in animals, plants, feed grains, and fermentation cultures [17,18]. A killer toxin provided by *P. anomala* AR_2016_ has no harmful effects on piglets [12]. *P. anomala* AR_2016_ grew best at pH 5.0 culture medium at 28 °C, 180 rpm, and tolerated 0.3% pig bile salts for 4 h, pH 2.0 artificial gastric juice for 1 h, and the artificial intestinal fluid for 4 h. The highest concentration of intestinal bile salt in animals is 0.3%. *Pichia anomala* AR_2016_ has been developed as a probiotic.

The serum contents of Glu and TP of pigs in the *P. anomala* and *C. utilis* groups were significantly higher than that of the CK group. The contents of Glu and TP in the serum of pigs reflect the degree of digestion and absorption of carbohydrate and protein substances in the feed [19]. The increase in daily intake and final weight in the *P. anomala* and the *C. utilis* group was associated with the increase in serum Glu and TP contents. The serum contents of AST, MDA, ALT, D-lactic acid, and endotoxins in pigs in the *P. anomala* and *C. utilis* groups were significantly lower than in the CK group. The serum contents of GSH-px, SOD, and T-AOC in the *P. anomala* and *C. utilis* groups were significantly higher than that of the CK group. The AST and ALT activity in the serum can reflect the degree of liver damage [20]. D-lactic acid and endotoxin are often used as indicators of intestinal permeability. This study indicated that the addition of yeast to the diet could improve the antioxidant capacity and alleviate the intestinal stress caused by the weaning of pigs.

When pigs are weaned, weaning stress and diet change have inhibitive effects on digestive enzyme activity because the intestinal digestive enzymes and digestive fluid secretion system, and micro-ecosystem have not yet formed [21]. The intestinal digestive enzymes (TRS, AMS, and LPS) in the *P. anomala* and *C. utilis* groups were higher than those in the CK group. It is speculated that the addition of active yeast to the diet may alleviate the shortage of intestinal digestive enzymes and improve the digestive and absorption functions of intestinal nutrients in pigs.

Small intestinal villi epithelial cells also dephosphorylate extracellular nucleotides, thereby reducing inflammation and regulating and maintaining the homeostasis of intestinal flora. When weaning pigs, the expression of intestinal alkaline phosphatase (ALP) is down-regulated, the LPS of symbiotic, pathogenic, and gram-negative bacteria increase the inflammatory process, and diarrhea incidence through TLR-4 activation. The addition of feed additives can improve the expression and activity of ALP and effectively reduce the incidence of diarrhea and promote intestinal health [22]. The occludin and ZO-1 concentrations in the jejunal mucosa in the *C. utilis* and *P. anomala* groups were significantly higher than CK group in this experiment. The tight junction protein occludin and ZO-1 are crucial to maintaining the integrity of intestinal epithelial cell structure, protect intestinal barrier function, and prevent bacterial endotoxin and toxic macromolecules from entering the body [23,24]. By extrapolation, the *P. anomala* group can inhibit the growth of harmful fungi by increasing the expression level of ALP and removing the LPS produced by harmful microorganisms, producing β-glucanase and organic acids, regulating the intestinal flora to promote intestinal health, reducing the mRNA expression level of APN to resist virus invasion, and thereby reduce the incidence of diarrhea.

In this study, the amino acid decarboxylase (HDC, LDC, and TDC) activity in the *P. anomala* and *C. utilis* groups was lower than in the CK group. Histidine decarboxylase could metabolize histidine to histamine and cause diarrhea [25]. The decrease in the incidence of diarrhea by live yeast was related to the decrease in the activity of histidine decarboxylase in intestinal microorganisms. Decarboxylase is mainly produced by bacteria, such as Enterobacteria or *Escherichia coli*. Biogenic amines are non-volatile amines generated by removing carboxyl groups under the action of amino acids decarboxylase. They are toxic substances produced in the putrefaction process of easily degraded feed and food. The higher the activity of microbial decarboxylase, the more biogenic amines will be generated, resulting in a low actual utilization rate of amino acids [26].

In this study, it was found that the dominant bacterial family in each treatment group were Bacteroidetes, Firmicutes, and Proteobacteria, of which Bacteroidetes and Firmicutes accounted for more than 90%, which was consistent with the results reported by Edward et al. [27]. The results of the current experiment show that the relative abundance of the flora can change:

(1) At the phylum level, the *P. anomala* group significantly increased the relative abundance of Bacteroides, significantly decreased the relative abundance of Proteobacteria, and the *C. utilis* group showed an upward trend of Bacteroides and a downward trend of Proteobacteria, but the difference was not significant. Bacteroidetes mainly produce propionic acid by fermentation, which has a maintenance effect on the intestinal barrier. Many Bacteroidetes are common bacteria in the animal intestinal tract, and the increase in the number of common bacteria may enhance the resistance to the invasion of foreign bacteria and maintain the balance of intestinal flora. Most of the Proteobacteria are pathogenic bacteria (such as *E. coli*, *Salmonella*). In this study, the addition of *P. anomala* significantly reduced the incidence of diarrhea, which may be related to the reduced relative abundance of Proteobacteria.

(2) Further analysis at the level of the order, family, and genus showed that the relative abundance of Platanaceae was the highest, and there was no significant difference between the groups. In the *P. anomala* group, the relative abundance of Vibrionaceae, Trichospirillaceae, and Lactobacillaceae was significantly increased, while the relative abundance of Rhodospirillales, Clostridiaceae, Vibrionaceae, and *Bacillus* spp. was significantly decreased. The relative abundance of Lachnospiraceae and Lactobacillaceae in the *C. utilis* group was significantly increased, while the relative abundance of Rhodospirillales, Clostridiaceae, and *Bacillus* spp. was significantly decreased.

Intestinal microflora maintains the stability of the intestinal environment. Changes in the microflora of weaned pigs due to diet changes, environment, and physiological state, lead to changes in the intestinal structure and function, which are extremely likely to lead to diseases [28]. A large number of microorganisms exist in the cecum, which is the main organ for microbial fermentation in pigs and other monogastric animals. Fermented carbohydrates can produce various volatile fatty acids, which play an important role in maintaining the intestinal barrier function of the host [29]. Animals cannot degrade dietary cellulose in the stomach and small intestine, but microorganism fermentation can degrade cellulose, hemicellulose, and pectin. *Prevotella* spp. can degrade cellulose, starch, and hemicellulose, while *Vibrio succinate* can decompose polysaccharides of plant cellulose and produce acetic acid, succinic acid, and other substances [30]. Lachnospiraceae can metabolize butyric acid substances [31]. Probiotics can increase the content of volatile fatty acids in the intestinal tract of pigs, thereby reducing pH and inhibiting the proliferation of pathogenic bacteria [29], which may be related to the fact that beneficial bacteria, such as yeast and lactic acid bacteria, can live in a slightly acidic environment, while pathogenic bacteria, such as *E. coli*, prefer a neutral or slightly alkaline environment. *Clostridium* is a genus of gram-positive bacteria causing human and animal intestinal disease, *Clostridium perfringens* and *C. difficile* can produce bacterial toxins and enterotoxins, causing the damaged intestinal mucosa seen in diarrhea [32]. *Clostridia* can cause clostridial enteritis, a major cause of diarrhea in pigs [33]. In this study, the significant decrease in the relative abundance of Clostridiaceae was one of the reasons for the reduction in the incidence of diarrhea, and the balance of microflora in the cecum of weaned pigs may be conducive to the healthy growth of pigs. Through alpha diversity analysis, it was found that the *P. anomala* group improved the abundance of the microflora and increased the species diversity, and the diversity and abundance of the microflora determined the structure and function of the microflora. Therefore, it can be speculated that live yeast can regulate the microflora, improve the proliferation of some beneficial bacteria, limit the proliferation of pathogenic bacteria, and maintain the intestinal health of weaned pigs.

In this study, it was found that oral administration of *P. anomala* AR_2016_ could reduce piglet diarrhea and improve piglet growth performance in 28-day-old weaned pigs. The *P. anomala* group was superior to the *C. utilis* group, and the results are consistent with previous studies [34,35]. Live yeast can regulate the gastrointestinal tract, maintain the body’s immunity, stimulate the immune system, alleviate the gastrointestinal injury caused by weaning stress, and enhance the body’s immunity to resist the invasion of pathogenic bacteria, thus reducing the incidence of diarrhea.

## 5. Conclusions

In this study, *P. anomala* was screened through traditional taxonomic and molecular biological classifications and identification methods, and was named AR_2016_. The strain of *P. anomala* AR_2016_ is resistant to acid, high bile salts, and artificial gastric and intestinal fluids. A killer toxin provided by *P. anomala* AR_2016_ has no harmful effects on piglets. *P. anomala* AR_2016_ the increase of average daily gain by 55.2% and the decrease of diarrhea incidence in piglets were related that *P. anomala* AR_2016_ improved digestive enzyme activity (trypsin, amylase, and lipase), enhanced intestinal barrier function (the ileal mucosa occludin, ZO-1, concentrations), decreased amino acids decarboxylase activity (HDC, LDC, and TDC), and intestinal digestive enzyme activity, reduced, and improved intestinal microflora of weaned pigs.

## 6. Patents

*Pichia anomala* and its fermentation culture and application. 2017. Chinese invention patent. ZL 201711298322.9.

## Figures and Tables

**Figure 1 animals-11-01179-f001:**
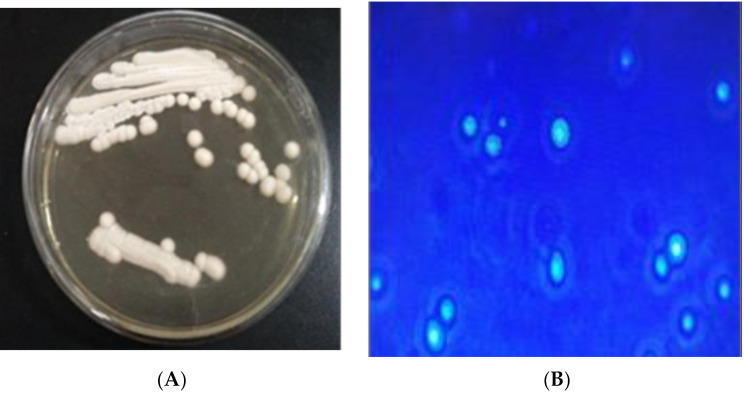
The colony and cell morphology of *P. anomala* AR_2016_. (**A**) Strain AR2016 was observed under the visual; (**B**) strain AR2016 was observed under the microscope.

**Figure 2 animals-11-01179-f002:**
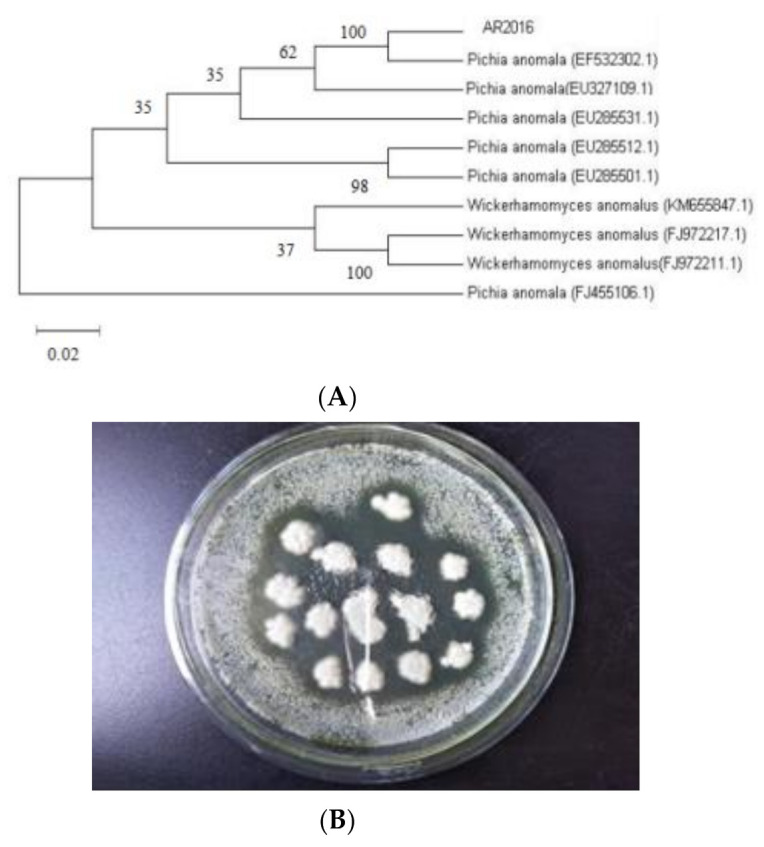
(**A**) Phylogenetic tree constructed of *P. anomala* AR_2016_ (**B**) the bacteriostatic effect of *P. anomala* AR_2016_ on *C. albicans*.

**Figure 3 animals-11-01179-f003:**
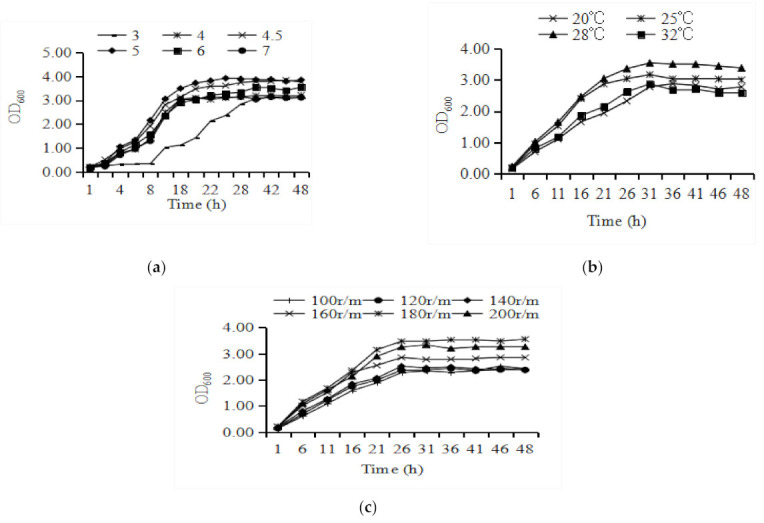
Optimization of Culture Conditions for *P. anomala* AR_2016_. (**a**) Effect of culture media pH on the growth of *P. anomala* AR_2016_; (**b**) effect of temperature on the growth of *P. anomala* AR_2016_; (**c**) effect of shaking speed on the growth of *P. anomala* AR_2016_.

**Figure 4 animals-11-01179-f004:**
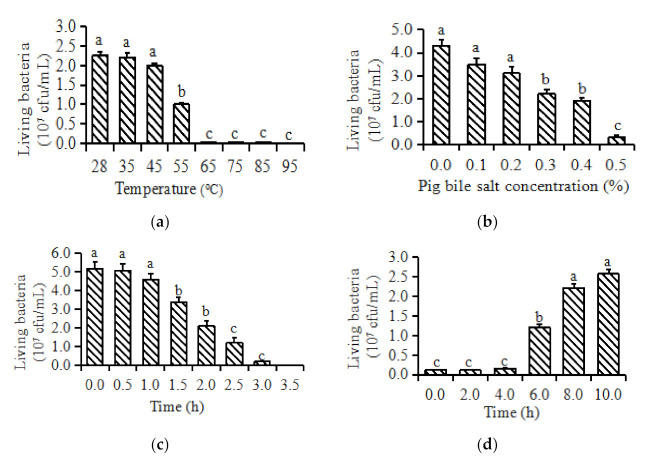
The Resistance of AR_2016._ (**a**) The effects of temperatures on the growth of *P. anomala* AR_2016_ (28 °C). The abscissa represents the temperature of bacterial culture; the ordinate represents the number of bacteria per milliliter of culture broth; (**b**) the effects of pig bile salts on the growth of *P. anomala* AR_2016_ (28 °C). The abscissa indicates the concentration of bile salt in the medium; the ordinate represents the number of bacteria per milliliter of culture broth; (**c**) the effects of simulated gastric fluid on the growth of *P. anomala* AR_2016_ (28 °C). The abscissa represents the time of bacteria in simulated gastric juice; the ordinate represents the number of bacteria per milliliter of culture broth; (**d**) the effects of simulated intestinal fluid on the growth of *P. anomala* AR_2016_ (28 °C). The abscissa represents the time of bacteria in the artificial intestinal fluid; the ordinate represents the number of bacteria per milliliter of culture broth.

**Figure 5 animals-11-01179-f005:**
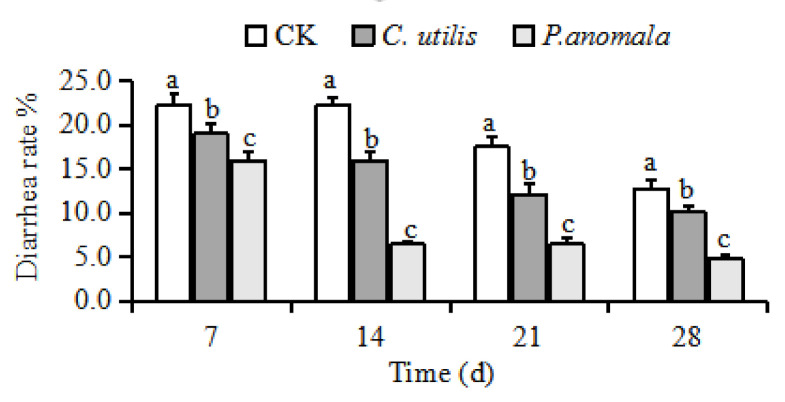
Effects of *C. utilis* and *P. anomala* AR_2016_ on diarrhea rate in weaned pigs (n = 10). Note: Piglets were orally administered 1 mL of 0.85% saline (CK), 1 mL 1 × 10^9^ cfu/mL *C. utilis* in 0.85% saline (*C. utilis*), or 1 mL 1 × 10^9^ cfu/mL *P. anomala* in 0.85% saline (*P. anomala*) once daily (12:00). ^abc^ Values with the same superscripts letter in the column means no significant difference (*p* > 0.05), values with different superscripts letter means significant difference (*p* < 0.05).

**Figure 6 animals-11-01179-f006:**
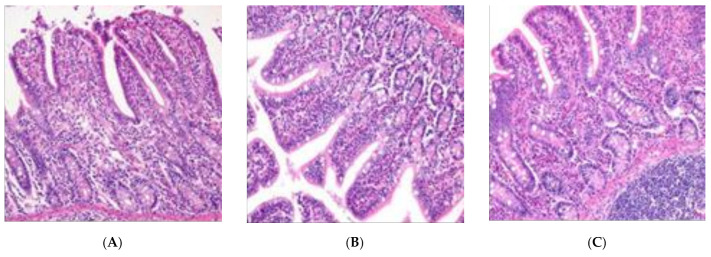
The effects of *C. utilis* and *P. anomala* AR_2016_ strain on villus height and crypt depth of ileum in weaned pigs (×400, (**A**) represents for the control group, (**B**) represents for *C*. *utilis* group, (**C**) represents for *P*. *anomala* group).

**Figure 7 animals-11-01179-f007:**
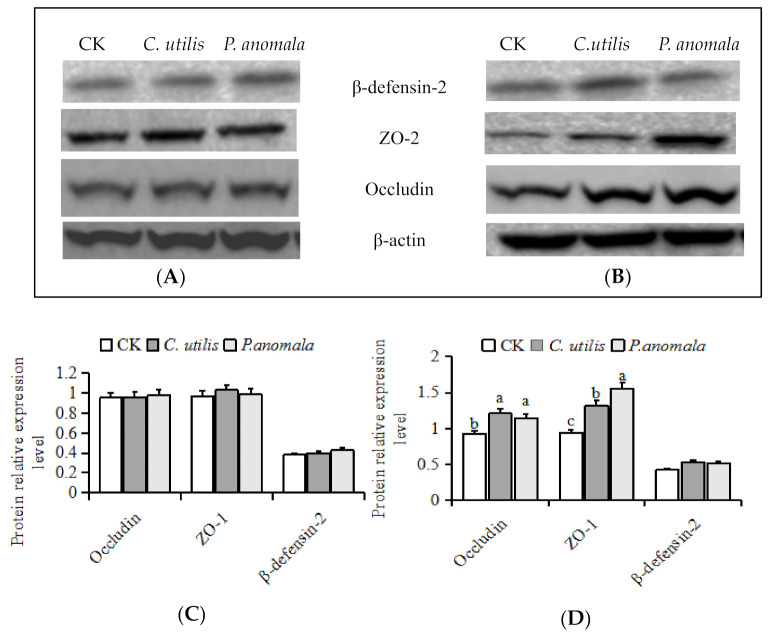
(**A**) The effects of *C. utilis* and *P. anomala* AR_2016_ on the intestinal mucosa occludin, ZO-1, and β-defensin-2 concentrations in jejunum (original western blot figures in Appendix A); (**B**) the effects of *C. utilis* and *P. anomala* AR_2016_ on the intestinal mucosa occludin, ZO-1, and β-defensin-2 concentrations in ileum (original western blot figures in Appendix A); (**C**) the jejunal mucosa occludin, ZO-1, and β-defensin-2 concentrations among three groups; (**D**) the ileal mucosa occludin, ZO-1, and β-defensin-2 concentrations among three groups. Note: Piglets were orally administered 1 mL of 0.85% saline (CK), 1 mL 1 × 10^9^ cfu/mL *C. utilis* in 0.85% saline (*C. utilis*), or 1 mL 1 × 10^9^ cfu/mL *P. anomala* in 0.85% saline (*P. anomala*) once daily (12:00). ^abc^ Values with the same or no superscripts letter in the bar means no significant difference (*p* > 0.05), values with different superscripts letter in the bar means significant difference (*p* < 0.05).

**Figure 8 animals-11-01179-f008:**
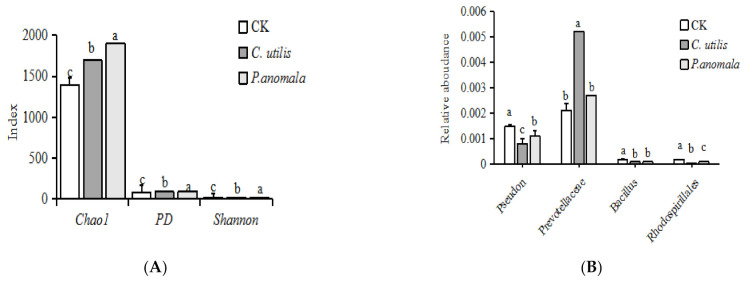
(**A**) Analysis of the alpha diversity; (**B**) main difference microbial floras. **Note:** Piglets were orally administered 1 mL of 0.85% saline (CK), 1 mL 1 × 10^9^ cfu/mL *C. utilis* in 0.85% saline (*C. utilis*), or 1 mL 1 × 10^9^ cfu/mL *P. anomala* in 0.85% saline (*P. anomala*) once daily (12:00). ^abc^ Values with the same superscripts letter in the column means no significant difference (*p* > 0.05), values with different superscripts letter means significant difference (*p* < 0.05).

**Table 1 animals-11-01179-t001:** The ingredients and nutritional compositions of diets (Dry Matter, basis) %.

Ingredients	Content	Nutritional Level ^3^	Content
Corn crude protein 8%	62.98	Digestive energy /(MJ/kg)	13.81
Soybean meal CP 43%	19.90	Crude protein	16.50
Fish meal CP 65%	2.40	Ca	0.73
Whey	5.65	Crude Fibre	2.60
Wheat Bran	5.08	Absorbable P	0.37
Soybean oil	0.85	Lysine (Lys)	1.27
Limestone	0.77	Methionine (Met)	0.37
Salt	0.30	Threonine (Thr)	0.75
CaHPO_4_	0.70	Tryptophan (Trp)	0.22
Sweetener	0.06		
Antioxidant	0.02		
Choline chloride	0.08		
Vitamin premix ^1^	0.08		
Trace mineral premix ^2^	0.30		
Threonine (Thr)	0.09		
Lysine Hydrochloride	0.31		
Methionine (Met)	0.09		
Tryptophan (Trp)	0.34		
Total	100.00		

^1^ Vitamin premix provided the following per kg of the diet: Vitamin A 2 017 IU, vitamin D 208 IU, vitamin E 14 IU, vitamin K 0.49 mg, pantothenic acid 10.1 mg, riboflavin 3.4 mg, folic acid 0.29 mg, nicotinic acid 29.1 mg, thiamine 1.1 mg, vitamin B_6_ 5.7 mg, biotin 0.06 mg, vitamin B_12_ 0.017 mg. ^2^ Trace mineral premix provided the following per kg of the diet: ZnSO_4_·7H_2_O 268 mg, FeSO_4_·7H_2_O 323.33 mg, MnSO_4_·H_2_O 11.54 mg, CuSO_4_·5H_2_O 22.86 mg, KI 14.21 mg, Na_2_SeO_3_ 28.37 mg. ^3^ Digestive energy was a calculated value, and the others were measured.

**Table 2 animals-11-01179-t002:** Primers used to amplify intestinal mucosa genes.

Genes	Product Size and Accession No.	Primer Sequence (5′→3′)	Tm Value
Glyceraldehyde-3-phosphate dehydrogenase	168 bp NM_001206359.1	F:AAGGTCGGAGTGAACGGATT R:GACTGTGCCGTGGAACTTG	59 °C
Alkaline phosphatase	199 bp XM_021097682.1	F:CCAAGCTCAGCAGACCCTAAA R:CAGGGCCACATAGGGAAACT	60 °C
Aminopeptidase N	114 bp NM_214277.1	F:GGACGATTGGGTCTTGCTGA R:GGGATGACCGACAGGTTTGT	60 °C
Toll-like receptors 2	112 bp NM_213761.1	F:CTCTGTCTTGTGACCCTGCT R:CCCACATAGGCGATCCTGTT	55 °C
Tumor necrosis factor-α	113 bp NM_214022.1	F:AACCTCAGATAAGCCCGTCG R:CTTTCAGCTTCACGCCGTTG	60 °C
Interleukin-10	124 bp NM_214041.1	F:CAGATGGGCGACTTGTTGCT R:CACTCTTGGCCTTCGGCATT	60 °C

**Table 3 animals-11-01179-t003:** Carbon sources assimilation of strain AR2016.

Carbon Source	Results	Carbon Source	Results
Galactose	+	Acetamide	+
Sucrose	+	Mannitol	+
Fiber disaccharide	+	Rhamnose	+
Raffinose	−	Melibiose	−
Sorbitol	+	Xylose	+
Melezitose	+	Ribose	+
Inositol	+	Glycerin	+
Trehalose	+	Arabinose	+

“+” said positive, “−” means negative.

**Table 4 animals-11-01179-t004:** Effects of *C. utilis* and *P. anomala* AR_2016_ strain on growth performance in weaned piglets (*n* = 10).

Items	Groups ^1^			SEM ^2^	*p*-Value
	CK	*C. utilis*	*P. anomala*		
Initial weight/kg	7.34	7.46	7.65	0.23	0.63
Final weight/kg	11.9 ^c^	13.8 ^b^	15.0 ^a^	0.29	<0.001
Average daily gain gain/(g/d)	163 ^b^	226 ^a^	262 ^a^	13.2	<0.001
Average daily feed intake/(g/d)	370 ^b^	450 ^a^	470 ^a^	17.5	0.01
Feed conversion rate	0.44 ^b^	0.50 ^a^	0.56 ^a^	0.10	0.04

^1^ Piglets were orally administered 1 mL of 0.85% saline (CK), 1 mL 1 × 10^9^ cfu/mL *C. utilis* in 0.85% saline (*C. utilis*), or 1 mL 1 × 10^9^ cfu/mL *P. anomala* in 0.85% saline (*P. anomala*) once daily (12:00). ^2^ Standard error of the mean. ^abc^ Values with the same superscripts letter in the column means no significant difference (*p* > 0.05), values with different superscripts letter means significant difference (*p* < 0.05).

**Table 5 animals-11-01179-t005:** The effects of *C. utilis* and *P. anomala* AR_2016_ on serum biochemical indicators in weaned piglets (*n* = 5).

Items	Groups ^1^			SEM ^2^	*p*-Value
	CK	*C. utilis*	*P. anomala*		
Glucose (mmol/L)	2.52 ^c^	5.19 ^a^	3.50 ^b^	0.20	<0.01
Total protein (g prot/L)	47.6 ^c^	57.6 ^a^	53.7 ^b^	0.68	<0.01
Total superoxide dismutase (U/mg L)	29.9 ^c^	54.4 ^b^	57.8 ^a^	1.06	<0.01
Total antioxidative capacity (U/mL)	1.17 ^c^	2.71 ^b^	22.30 ^a^	0.40	<0.01
Glutathione peroxidase (U/mg L)	482 ^c^	518 ^b^	564 ^a^	1.00	<0.01
Malondialdehyde (nmoL/L)	10.36 ^a^	2.54 ^b^	2.28 ^b^	0.155	<0.01
D-lactic acid (ng/μL)	72.40 ^a^	39.46 ^b^	38.85 ^b^	1.072	<0.01
Endotoxin (EU/mL)	0.182 ^a^	0.142 ^b^	0.112 ^c^	0.005	<0.01

^1^ Piglets were orally administered 1 mL of 0.85% saline (CK), 1 mL 1 × 10^9^ cfu/mL *C. utilis* in 0.85% saline (*C. utilis*), or 1 mL 1 × 10^9^ cfu/mL *P. anomala* in 0.85% saline (*P. anomala*) once daily (12:00). ^2^ Standard error of the mean. ^abc^ Values with the same superscripts letter in the column means no significant difference (*p* > 0.05), values with different superscripts letter means significant difference (*p* < 0.05).

**Table 6 animals-11-01179-t006:** The effects of *C. utilis* and *P. anomala* AR_2016_ on digestive enzyme activity, villus height, and crypt depth in weaned piglets (*n* = 5).

Items	Groups ^1^			SEM ^2^	*p*-Value
	CK	*C. utilis*	*P. anomala*		
Jejunum					
Trypsin (U/mg prot)	26.90 ^c^	28.40 ^b^	31.30 ^a^	0.304	<0.01
Amylase (U/mg prot)	0.05 ^c^	0.07 ^b^	0.08 ^a^	0.003	<0.01
Lipase (U/mg prot)	144 ^c^	346 ^a^	323 ^b^	4.327	<0.01
Ileum					
Trypsin (U/mg prot)	29.53 ^b^	37.13 ^a^	39.25 ^a^	0.757	<0.01
Amylase (U/mg prot)	0.09	0.09	0.09	0.003	0.93
Lipase (U/mg prot)	190 ^c^	340 ^a^	307 ^b^	6.986	<0.01
Villus height /(μm)	253 ^b^	253 ^b^	354 ^a^	8.10	0.01
Crypt depth /(μm)	182 ^a^	147 ^b^	176 ^a^	4.46	0.02
Villus height/crypt depth	1.39 ^c^	1.73 ^b^	2.02 ^a^	0.04	<0.01

^1^ Piglets were orally administered 1 mL of 0.85% saline (CK), 1 mL 1 × 10^9^ cfu/mL *C. utilis* in 0.85% saline (*C. utilis*), or 1 mL 1 × 10^9^ cfu/mL *P. anomala* in 0.85% saline (*P. anomala*) once daily (12:00). ^2^ Standard error of the mean. ^abc^ Values with the same superscripts letter in the column means no significant difference (*p* > 0.05), values with different superscripts letter means significant difference (*p* < 0.05).

**Table 7 animals-11-01179-t007:** The effects of *C. utilis* and *P. anomala* AR_2016_ strain on the mRNA level of the intestinal mucosa genes of weaned piglets (*n* = 5).

Items	Groups ^1^			SEM ^2^	*p*-Value
	CK	*C. utilis*	*P. anomala*		
Jejunum					
Alkaline phosphatase	0.62 ^b^	0.78 ^a^	0.83 ^a^	0.04	0.01
Aminopeptidase N	0.24	0.23	0.19	0.02	0.17
Toll-like receptors 2	0.31 ^b^	0.28 ^b^	0.48 ^a^	0.02	<0.01
Tumor necrosis factor-α	0.21 ^c^	0.55 ^a^	0.37 ^b^	0.03	<0.01
Interleukin-10	0.96	1.28	1.21	0.07	0.05
Ileum					
Alkaline phosphatase	0.42 ^b^	0.51 ^a^	0.55 ^a^	0.02	0.01
Aminopeptidase N	0.26 ^a^	0.19 ^b^	0.14 ^b^	0.02	0.01
Toll-like receptors 2	0.21	0.18	0.29	0.03	0.09
Tumor necrosis factor-α	0.19 ^c^	0.46 ^a^	0.36 ^b^	0.02	<0.01
Interleukin-10	0.82	1.15	1.09	0.05	0.05

^1^ Piglets were orally administered 1 mL of 0.85% saline (CK), 1 mL 1 × 10^9^ cfu/mL *C. utilis* in 0.85% saline (*C. utilis*), or 1 mL 1 × 10^9^ cfu/mL *P. anomala* in 0.85% saline (*P. anomala*) once daily (12:00). ^2^ Standard error of the mean. ^abc^ Values with the same superscripts letter in the column means no significant difference (*p* > 0.05), values with different superscripts letter means significant difference (*p* < 0.05).

**Table 8 animals-11-01179-t008:** The effects of *C. utilis* and *P. anomala* AR2016 strain on Amino acid decarboxylase level of the intestine of weaned piglets (*n* = 5).

Items	Groups ^1^			SEM ^2^	*p*-Value
	CK	*C. utilis*	*P. anomala*		
Jejunum (U/mg)					
Arginine decarboxylase	1.42	1.38	1.37	0.050	0.76
Histidine decarboxylase	1.61 ^a^	1.45 ^b^	1.40 ^b^	0.037	<0.01
Lysine decarboxylase	1.44	1.41	1.35	0.029	0.17
Methionine decarboxylase	1.37	1.33	1.33	0.032	0.79
tryptophan decarboxylase	0.57	0.48	0.49	0.027	0.08
Ileum (U/mg)					
Arginine decarboxylase	1.64	1.61	1.65	0.014	0.15
Histidine decarboxylase	1.60 ^a^	1.44 ^b^	1.31 ^b^	0.019	<0.01
Lysine decarboxylase	1.53 ^a^	1.35 ^b^	1.03 ^c^	0.008	<0.01
Methionine decarboxylase	1.22	1.23	1.23	0.059	0.99
tryptophan decarboxylase	0.40 ^a^	0.33 ^b^	0.26 ^c^	0.014	<0.01

^1^ Piglets were orally administered 1 mL of 0.85% saline (CK), 1 mL 1 × 10^9^ cfu/mL *C. utilis* in 0.85% saline (*C. utilis*), or 1 mL 1 × 10^9^ cfu/mL *P. anomala* in 0.85% saline (*P. anomala*) once daily (12:00). ^2^ Standard error of the mean. ^abc^ Values with the same superscripts letter in the column means no significant difference (*p* > 0.05), values with different superscripts letter means significant difference (*p* < 0.05).

**Table 9 animals-11-01179-t009:** The cecum microflora abundance at phylum level and alpha diversity (*n* = 3).

Items	Groups ^1^			SEM ^2^	*p*-Value
	CK	*C. utilis*	*P. anomala*		
*Bacteroidetes*	59.90 ^b^	62.10 ^b^	66.50 ^a^	1.04	0.01
*Prevotellaceae*	48.96	50.63	52.94	0.97	0.07
*Firmicutes*	26.50	24.20	27.70	0.92	0.09
*Proteobacteria*	11.00 ^a^	10.70 ^a^	3.40 ^b^	0.30	<0.01
*Lachnospiraceae*	10.92^c^	12.22 ^b^	14.22 ^a^	0.27	<0.01
*Bacteroidaceae*	7.58	6.40	6.72	0.40	0.18
*Actinobacteria*	5.40 ^a^	1.20 ^b^	1.10 ^b^	0.15	<0.01
*Ruminococcaceae*	4.97	5.34	5.53	0.40	0.62
*Clostridiaceae*	4.75 ^a^	1.08 ^b^	1.40 ^b^	0.18	<0.01
*Veillonellaceae*	3.09	3.50	3.66	0.22	0.25
*Lactobacillaceae*	2.89 ^a^	0.38 ^c^	1.29 ^b^	0.22	<0.01
*Succinivibrionaceae*	1.72 ^b^	1.96 ^ab^	2.41 ^a^	0.14	0.04
*Acidaminococcaceae*	0.36	0.35	0.33	0.10	0.98
*Desulfovibrionaceae*	0.31 ^a^	0.05 ^b^	0.03 ^b^	0.02	<0.01
*Enterobacteriaceae*	0.20	0.11	0.33	0.08	0.28
*Campylobacteraceae*	0.07 ^a^	0.08 ^a^	0.03 ^b^	0.01	0.02

^1^ Piglets were orally administered 1 mL of 0.85% saline (CK), 1 mL 1 × 10^9^ cfu/mL *C. utilis* in 0.85% saline (*C. utilis*), or 1 mL 1 × 10^9^ cfu/mL *P. anomala* in 0.85% saline (*P. anomala*) once daily (12:00). ^2^ Standard error of the mean. ^abc^ Values with the same superscripts letter in the column means no significant difference (*p* > 0.05), values with different superscripts letter means significant difference (*p* < 0.05).

## Data Availability

The data presented in this study are available on request from the corresponding author. The data are not publicly available due to the regulations of National Natural Science Foundation of China.

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
