# Peer review of "Isolation, Identification and Function of Pichia anomala AR2016 and Its Effects on the Growth and Health of Weaned Pigs"

_animals, 2021, doi:10.3390/ani11041179_

Round 1
Reviewer 1 Report
The authors need to address why there was no basal blood sample collected from the animals. Line 176
How were the pigs selected for post-mortem sampling? Line 181
Author Response
Section : "Responses to the Comments by reviewer 1"
Comment 1: The authors need to address why there was no basal blood sample collected from the animals. Line 176
Authors'responses and locations of the revisions: There is address error. basal blood sample were collected from piglets in CK group. We have corrected it. (Please see lines 210-211 in the revised version, Red letter marked).
Comment 2: How were the pigs selected for post-mortem sampling? Line 181
Authors'responses and locations of the revisions: Thanks. Owing to your suggestion, we have added the conditions for sampling pigs. (Please see lines 215-217 in the revised version, Red letter marked).
Reviewer 2 Report
Review
The paper "Isolation, identification, and function of Pichia anomalaAR20162and its effects on the growth and health of weaned pigs" by the authors: Yajun Ma, Zhihong Sun, Yan Zeng, Ping Hu, Weizhong Sun, Yubo Liu, Hong Hu, Zebin Rao and Zhiru Tang, present a study on Pichia anomala, which is a simple pseudo mycelium that produces shoots and reproduces vegetative somatic cells. Candida albicans has been shown to show resistance to gastric juice and bile in piglets weaned from sows.
Evaluating positively the results obtained, however, it is necessary to point out a number of drawbacks:
- In the introduction very few sources of literature on the topic under study are given;
- Section "Material and methods of research" needs to be finalized;
- In the section "Research results" the obtained results should be presented more clearly, namely M±m for each group studied, because in this case we can check the reliability of the difference in the studied indicators;
- the authors point out that using the strain P.anomala AR 2016 increases the activity of histidine decarboxylase, which affects the histidine content in the body, but no information on the normative histidine content is provided. Would an increase in histidine have a negative effect on the body?
- In the conclusions there is no specific result of the work performed, but only general phrases that increased and improved, but by how much?
After completion, the material can be recommended for publication.
Author Response
Section: "Responses to the Comments by reviewer 2"
Comment 1: In the introduction very few sources of literature on the topic under study are given
Authors'responses and locations of the revisions: Thanks. Owing to your suggestion, (Please see lines “introduction” in the revised version, Red letter marked).
Comment 2: Section "Material and methods of research" needs to be finalized
Authors'responses and locations of the revisions: Thanks. Owing to your suggestion,We finalized Section "Material and Methods" (in the revised version, Red letter marked).
Comment 3: In the section "Research results" the obtained results should be presented more clearly, namely M±m for each group studied, because in this case we can check the reliability of the difference in the studied indicators
Authors'responses and locations of the revisions: Thanks. We have provided the SEM (Standard error of the mean) in tables.
Comment 4: the authors point out that using the strain P. anomala AR 2016 increases the activity of histidine decarboxylase, which affects the histidine content in the body, but no information on the normative histidine content is provided. Would an increase in histidine have a negative effect on the body?
Authors'responses and locations of the revisions: Thanks. Owing to your suggestion, in this experiment, the bacteria reduced the activity of histidine decarboxylase in the intestinal microbes, which is beneficial to reduce the diarrhea rate of weaned piglets, Histidine decarboxylase can metabolize histidine to histamine and cause diarrhea. (Please see Lines 611-618 in the revised version)Gallardo, P., Izquierdo, M., Vidal, R.M., Soto, F., Ossa, J.C., Farfan, M.J., 2020. Gut Microbiota-Metabolome Changes in Children With Diarrhea by Diarrheagenic E. coli. Frontiers in Cellular and Infection Microbiology 10. doi:10.3389/fcimb.2020.00485)
Comment 5: In the conclusions there is no specific result of the work performed, but only general phrases that increased and improved, but by how much?
Authors'responses and locations of the revisions: Thanks. Owing to your suggestion, we have provide specific result description. (Please see line 666-692 in the revised version, Red letter marked).
Reviewer 3 Report
After evaluating the article, we are positive to the publishing if the following topics are considered:
Line 18 mentions CK sterile water and then in the text as in line 153, mentions saline solution 0.85%. Review the text;
Line 82, review “thre”;
Line 141-148, duplicated text;
Figure two (2 A), figure is unreadable;
Figure four (4), what is the meaning of the letters in the bars?
Table one (1), explain better the abbreviations of the columns of the different aminoacids;
Figure four (4), which temperature were the tests performed?
Figure seven (7), what is the statistic difference between the bars of figure C and D?
Author Response
Section: "Responses to the Comments by reviewer 3"
Comment 1: Line 18 mentions CK sterile water and then in the text as in line 153, mentions saline solution 0.85%. Review the text;
Authors'responses and locations of the revisions: Thanks. Owing to your suggestion, this is a mistake, the control group orally administered 0.85% saline. (Please see lines31 in the revised version, Red letter marked).
Comment 2: Line 82, review “thre”
Authors'responses and locations of the revisions: Thanks. Owing to your suggestion, we have corrected the wrong word and changed "thre day old" to "three-day-old". (Please see line 101 in the revised version, Red letter marked).
Comment 3: Line 141-148, duplicated text
Authors'responses and locations of the revisions: Thanks. Owing to your suggestion, we double-checked the article and removed duplicate parts. (Please see the revised version)
Comment 4: Figure two (2 A), figure is unreadable
Authors'responses and locations of the revisions: we have provide more clear figure 2A (Please see figure 2A in the revised version)
Comment 5: Figure four (4), what is the meaning of the letters in the bars?
Authors'responses and locations of the revisions: Thanks. Owing to your suggestion, we have added an explanation of the text in the chart in the legend. (Please see figure 4 the revised version)
Comment 6: Table one (1), explain better the abbreviations of the columns of the different amino acids
Authors'responses and locations of the revisions: Thanks. Owing to your suggestion, we changed the abbreviation to the full name (Please see Table 1 in the revised version)
Comment 7: Figure four (4), which temperature were the tests performed?
Authors'responses and locations of the revisions: Thanks. Line 110 shows that the test was carried out at 28°C meantime we have increased the test temperature in the legend. (Please see figure 4 in the revised version)
Comment 8: Figure seven (7), what is the statistic difference between the bars of figure C and D?
Authors'responses and locations of the revisions: Thanks. Owing to your suggestion, we have add “ abc Values with the same or no superscripts letter in the bar means no significant difference (P>0.05), values with different superscripts letter in the bar means significant difference (P<0.05) the statistic difference between the bars of figure” (Please see Figure seven in the revised version)
Reviewer 4 Report
The authors have performed a very thorough piece of work on this novel probiotic. All techniques have been described in sufficient detail.
The manuscript is certainly of good quality, yet with one major comment that needs to be adressed: the discussion is cut in pieces per topic whereas it is expected that a discussion presents an integrated view on the results. This makes the present discussion fairly superficial.
Some detailed comments:
51: "piglets"
80: "serially"
83: "was photographed"
108: "Candida"
173: "ratio" (change throughout manuscript)
188: no capitals needed
224: it is not according to international guidelines (e.g. MIQE) to only have one reference gene. The authors should show the stability test of GAPDH and justify why only one gene was used.
271: Duncan is an allowed post-hoc test, but it is very "permissive", meaning a tendency to show false-positive results. The authors do not need to change this, but I would advise to use Tukey or Scheffé or so in the future in order to have more balanced conclusions.
Table 5: The authors should discuss why glucose was different. This is an example of results that have not been discussed (in an integrated way). For instance, is this an effect of altered digestive enzyme activity, altered gut morphology, altered post-absorptive glucose clearance, or combinations?
In the beginning of the Results, Discussion and Conclusions sections, there are some lines that are copied from guidelines, and should be removed.
Author Response
Comment 1: The authors have performed a very thorough piece of work on this novel probiotic. All techniques have been described in sufficient detail. The manuscript is certainly of good quality, yet with one major comment that needs to be adressed: the discussion is cut in pieces per topic whereas it is expected that a discussion presents an integrated view on the results. This makes the present discussion fairly superficial.
Authors'responses and locations of the revisions: Thanks. Owing to your suggestion, we have revised the section “Discussion” in the revised version, Red letter marked).
Comment 2: Some detailed comments:51: "piglets"80: "serially"83: "was photographed"108: "Candida"173: "ratio" (change throughout manuscript)188: no capitals needed
Authors'responses and locations of the revisions: Thanks. Owing to your suggestion, we have corrected all error in the revised version, Red letter marked).
Comment 3: 224: it is not according to international guidelines (e.g. MIQE) to only have one reference gene. The authors should show the stability test of GAPDH and justify why only one gene was used.
Authors'responses and locations of the revisions: we have checked the stability test of GAPDH in the previous study. (Tang, Z.R.; Deng, H.; Sun, W.Z.; Zhang, X.L.; Zhang, Z. Study on the mechanism of orally administered probiotics Escherichia coli Nissle 1917 regulating intestine barrier in weaned piglets. Anim. Feed. Sci. Technol. 2014, 45, 78-86. doi: 10.11843/j.issn.0366-6964.2014.01.011)
Comment 4: 271: Duncan is an allowed post-hoc test, but it is very "permissive", meaning a tendency to show false-positive results. The authors do not need to change this, but I would advise to use Tukey or Scheffé or so in the future in order to have more balanced conclusions.
Authors'responses and locations of the revisions: Thanks.
Comment5: Table 5: The authors should discuss why glucose was different. This is an example of results that have not been discussed (in an integrated way). For instance, is this an effect of altered digestive enzyme activity, altered gut morphology, altered post-absorptive glucose clearance, or combinations?
Comment 6: In the beginning of the Results, Discussion and Conclusions sections, there are some lines that are copied from guidelines, and should be removed.
Authors'responses and locations of the revisions: Thanks. We have removed the lines.
Round 2
Reviewer 2 Report
I believe that the presented article can be published. She will make a significant contribution to the study of various strains of bacteria, including P. anomala, in the formation of microflora of the gastrointestinal tract, which has a positive effect on the productivity of pigs.